# Enhanced peer-group strategies to support the prevention of mother-to-child HIV transmission leads to increased retention in care in Uganda: A randomized controlled trial

Alexander Amone[1]*, Grace Gabagaya[1], Priscilla Wavamunno[1], Gordon Rukundo[1], Joyce Namale-Matovu[1], Samuel S. Malamba[2], Irene Lubega[1], Jaco Homsy[3], Rachel King[3], Clemensia Nakabiito[1], Zikulah Namukwaya[1], Mary Glenn Fowler[1,4], Philippa Musoke[1,5]

1 Makerere University-Johns Hopkins University Research Collaboration, Kampala, Uganda, 2 Uganda Virus Research Institute, Entebbe, Uganda, 3 Institute for Global Health Sciences, University of California, San Francisco, San Francisco, CA, United States of America, 4 Department of Pathology, Johns Hopkins University School of Medicine, Baltimore, MD, United States of America, 5 Department of Pediatrics and Child Health, Makerere University College of Health Sciences, Kampala, Uganda

☯ These authors contributed equally to this work.
* aamone@mujhu.org

## Abstract

### Introduction

Despite the scale-up of Option B+, long-term retention of women in HIV care during pregnancy and the postpartum period remains an important challenge. We compared adherence to clinic appointments and antiretroviral therapy (ART) at 6 weeks, 6, and and 24 months postpartum among pregnant women living with HIV and initiating Option B+. Women were randomized to a peer group support, community-based drug distribution and income-generating intervention called "Friends for Life Circles" (FLCs) versus the standard of care (SOC). Our secondary outcome was infant HIV status and HIV-free survival at 6 weeks and 18 months postpartum.

### Methods

Between 16 May 2016 and 12 September 2017, 540 ART-naïve pregnant women living with HIV at urban and rural health facilities in Uganda were enrolled in the study at any gestational age. Participants were randomized 1:1 to the unblinded FLC intervention or SOC at enrolment and assessed for adherence to the prevention of mother-to-child HIV transmission (PMTCT) clinic appointments at 6 weeks, 12, and 24 months postpartum, self-reported adherence to ART at 6 weeks, 6 and 24 months postpartum and supported by plasma HIV-1 RNA viral load (VL) measured at the same time points, retention in care through the end of study, and HIV status and HIV-free survival of infants at 18 months postpartum. The FLC groups were formed during pregnancy within 4 months of enrollment and held monthly meetings in their communites, and were followed up until the last group participant reached

**Data Availability Statement:** All relevant data are within the paper and its Supporting Information files.

**Funding:** PM received funding from NIH/Eunice Kennedy Shriver National Institute of Child Health and Human Development (NICHD) grant # IR01HD080476-01. The funders had no role in study design, data collection and analysis, decision to publish, or preparation of the manuscript.

**Competing interests:** PM received funding from NIH/Eunice Kennedy Shriver National Institute of Child Health and Human Development (NICHD) grant # IR01HD080476-01.

24 months post delivery. We used Log-rank and Chi-Square p-values to test the equality of Kaplan-Meier survival probabilities and hazard rates (HR) for failure to retain in care for any reason by study arm.

## Results

There was no significant difference in adherence to PMTCT clinic visits or to ART or in median viral loads between FLC and SOC arms at any follow-up time points. Retention in care through the end of study was high in both arms but significantly higher among participants randomized to FLC (86.7%) compared to SOC (79.3%), p = 0.022. The adjusted HR of visit dropout was 2.4 times greater among participants randomized to SOC compared to FLC (aHR = 2.363, 95% CI: 1.199–4.656, p = 0.013). Median VL remained < 400 copies/ml in both arms at 6 weeks, 6, and 24 months postpartum. Eight of the 431 infants tested at 18 months were HIV positive (1.9%), however, this was not statistically different among mothers enrolled in the FLC arm compared to those in the SOC arm. At 18 months, HIV-free survival of children born to mothers in the FLC arm was significantly higher than that of children born to mothers in the SOC arm.

## Conclusions

Our findings suggest that programmatic interventions that provide group support, community-based ART distribution, and income-generation activities may contribute to retention in PMTCT care, HIV-free survival of children born to women living with HIV, and ultimately, to the elimination of mother-to-child HIV transmission (EMTCT).

## Trial registration

NCT02515370 (04/08/2015) on ClinicalTrials.gov.

## Introduction

Prevention of mother-to-child HIV transmission (PMTCT) remains a pillar in the global fight against HIV/AIDS. "Option B+" for PMTCT recommends lifelong triple ART for all pregnant and breastfeeding women living with HIV regardless of immune status. Option B+ also recommends protecting these women's newborns with daily Nevirapine or Zidovudine from birth through 4–6 weeks of age, regardless of the infant feeding method [1].

In April 2012, the World Health Organisation (WHO) recommended Option B+ as the PMTCT treatment option which would provide the highest clinical benefits and programmatic advantages for both the care and treatment of mothers living with HIV and perinatal HIV prevention [1]. Compared to previous proven PMTCT regimens, Option B+ was easier to deliver and more effective in reducing mother-to-child HIV transmission (MTCT) [2–4]. Based on these recommendations, several low-income countries including Uganda adopted Option B + as the standard of care with the aim to eliminate the MTCT of HIV.

Despite this progress and the proven efficacy of Option B+ [5], poor adherence to lifelong ART and low retention in care of pregnant and postpartum women living with HIV have remained an important programmatic challenge to the implementation of Option B+ especially in low-resource settings [6–11]. A scoping literature review of studies done between 2000–

2020 identified barriers to taking ART among pregnant women that included financial constraints limiting access to food and transport, and side effects of the therapy [12]. Additionally, high dropout rates from PMTCT care have been documented after delivery and at two years of follow-up among mothers living with HIV [13–15], suggesting that service integration and linking mothers to routine ART services were important determinants of retention in care. These findings highlighted the critical need for innovative interventions to promote retention in HIV care and adherence to ART for the successful implementation of option B$^+$.

In Uganda, adherence counseling by trained clinic-based midwives and the use of "buddies" as well as family involvement are recommended by the Ministry of Health (MOH) to support maternal ART adherence and retention in care [16]. However, there is limited focus on the use of community-based support interventions. Studies in other resource-limited settings have shown better PMTCT intervention adherence with the use of community-based counseling, support groups, and home visits [17–20]. Additionally, studies in Kenya have demonstrated that peer support from community-based mentor mothers to improve ART adherence and retention in care was well accepted among women living with HIV [21, 22].

Given these challenges, we conducted a randomized controlled trial (RCT) to compare adherence to PMTCT clinic appointments and ART at different time points within 24 months of postpartum follow-up among women started on ART while pregnant. The women were randomized to an enhanced peer group support intervention including community-based drug distribution and income-generating activities called "Friends for Life Circles" (FLCs) or to the MOH standard of care (SOC).

## Materials and methods

We conducted a non-blinded RCT between 16 May 2016 and 26 May 2020 at Mulago National Referral Hospital and two government health centers (HCs) in Kampala, the capital city of Uganda; and at rural Mityana District Hospital (74 km northwest of Kampala), as well as five HCs in Mityana district. Participants were recruited from 16 May 2016 to 12 September 2017 and follow-up was completed on 26 May 2020. The selected HCs represent different levels of maternal health care offered in Uganda. The study was implemented and coordinated by the Makerere University–Johns Hopkins University (MUJHU) Research Collaboration based at Mulago Hospital in Kampala and was registered on ClinicalTrials.gov with Trial registration: NCT02515370 (04/08/2015).

### Enrollment and randomization procedures

We enrolled pregnant women living with HIV at any gestation age initiated on PMTCT Option B+ who were: ≥ 18 years old, confirmed pregnant by urine dipstick, confirmed HIV-positive by rapid HIV testing as per National PMTCT Guidelines, self-declared ART-naïve. Additional inclusion criteria included: residing within 20 km of the study clinic, not planning to move out of the clinic catchment area within 2 years, agreeing to be home visited, and providing written informed consent.

Pregnant women at any gestational age newly diagonised HIV-positive by testing through the routine National PMTCT Program and starting Option B+ were referred to study counselors by the clinic PMTCT counselors for prescreening. If eligible, they were started on once daily Option B+ ART within 30 days of enrolment into the study and were scheduled to return to the study clinics for consent procedures within one month of starting Option B+ ART.

HIV rapid testing followed the national and WHO algorithms [23] using three sequential antibody rapid tests: Abbott Determine Rapid HIV 1 Test, Abbott Laboratories, USA; Stat Pak

Rapid HIV-1 Test, Chembio Diagnostic Systems, USA; and Unigold HIV Rapid Test, Trinity Biotech, Ireland.

At the time of enrolment, eligible and consenting women were randomized 1:1 irrespective of the study sites to SOC control arm or FLC intervention arm. The randomization list was computer-generated by the data manager based at MUJHU using random-sized block groups with block sizes ranging between 2–8 that included consecutive intervention numbers with corresponding random intervention assignments. Each pregnant woman was assigned a unique participant identification number and authors had no access to information that could identify individual participants during or after data collection.

After randomization, participants were assessed for socioeconomic status, ARV drug adherence, and stigma. A questionnaire addressing participants' experiences with stigmatization and discrimination at family and community levels was administered at baseline, 12 months post-enrolment, and the end of the study. An additional needs assessment questionnaire was administered to participants enrolled in the FLC arm at enrolment, 1-year post enrolment, and every 6 months thereafter until the end of the study. This was to assess individuals' achievements and to document challenges related to group activities and participation in income-generating activities (IGAs) and their benefits, skills acquired, and training needs.

## FLC intervention and SOC controls

The FLC intervention (hereafter 'FLC') included community-based enhanced peer group support, IGAs, and community-based drug distribution. Participants in the FLC arm formed dynamic peer support groups within four months of enrollment based on women's home addresses documented by Geographical Positioning System (GPS). Groups consisted of 8–10 participants who attended monthly meetings in their communities. Sustainable livelihood study assistants and study counselors participated in and documented FLC group activities. Each FLC group was supported by one peer mother volunteer who had similar social backgrounds and/or life experiences as FLC participants.

During monthly FLC group meetings, study counselors distributed ART and Cotrimoxazole and offered psychosocial support on follow-up clinic visits, drug adherence through individual and group counseling while emphasizing peer support among participants. Study counselors also offered infant feeding counseling with emphasis on the importance of breastfeeding on demand, complementary feeding from 6 months onword, and hygiene in preparing infant feeds.

FLC groups also offered skills building through a variety of IGAs such as goat rearing, bakery, crafts making, liquid soap making, bookbinding, and charcoal briquette making. These were chosen by the participants and implemented as a group or as individuals with the guidance of a study IGA Advisor. All FLC groups were linked to local government bodies for sustainability, registration, and access to Community Development Driven grants funded by the District Local Government to support the implementation of livelihood projects for community-based groups. The FLC groups were followed up until the last participant from the group reached 24 months post delivery and were encouraged to continue meeting monthly and implementing their chosen IGA beyond the study.

SOC arm participants received counseling from clinic PMTCT counselors and collected their drug refills on an individual basis from the PMTCT clinics where they were enrolled as per MOH guidelines [16]. SOC arm participants participated voluntarily in family support groups whenever available at their respective health facilities as recommended by the MOH [16].

## Follow-up procedures

At each study site, all study participants were followed at a dedicated clinic for all scheduled and unscheduled visits and were given transport reimbursement for scheduled study visits. Scheduled visits included a first visit within the initial four weeks post-enrolment, then monthly visits thereafter until delivery, at delivery, at 6 and 14 weeks postpartum, and quarterly thereafter from 6 months through 24 months postpartum (end of follow-up). At each of these visits, FLC participants were counseled by study counselors on drug adherence and family planning while SOC participants were counseled by PMTCT clinic counselors. Study counselors administered questionnaires on ART adherence, stigma, and socio-economic status to all participants at each scheduled clinic visit.

## Participants were terminated from the study if they missed four or more consecutive scheduled study visits, withdrew consent, relocated outside the study area, or died. Infant procedures

All infants born to study participants were enrolled in the study and received daily Nevirapine syrup from birth up to 6 weeks of age regardless of the infant feeding method. Infants were tested routinely at 6 weeks of age for HIV by DNA PCR and at 18 months of age by rapid HIV antibody test as part of the National Early Infant HIV Diagnosis Program [23].

## Outcomes and measures

The primary outcomes of interest included adherence to PMTCT clinic appointments at 6 weeks, 12, and 24 months postpartum, adherence to Option B+ ART at 6 weeks, and 6 and 24 months postpartum, and retention in care at the end of study follow up.

Complete adherence to visits was defined as having attended the 4 scheduled quarterly visits at the end of each of the first and second years of postpartum follow-up as defined by the MOH [16]. Scheduled visits were determined to have been adhered to based on a completed visit within a 3-week window on either side of the scheduled visit date.

Retention in care was defined as the proportion of participants who were not terminated from the study before the end of follow-up (24 months postpartum). Retention in care was thus calculated as 1 minus the proportion of women terminated for any such reason before the end of follow-up. Causes for termination included participant relocation out of the study area, withdrawal from the study, loss to follow-up, miscarriage, referral outside the study facility, and death. The study defined loss to follow-up as not attending 3 consecutive scheduled study visits.

We compared maternal ART adherence using self-report at 6 weeks and 6 and 24 months postpartum among all study participants. All participants were asked to self-assess the number of complete ART doses they missed in the past three days. Self-reported ART adherence was then calculated as 1 minus the proportion of self-reported doses missed in the past 3 days.

We validated self-reports by comparing them with VL measurements for all participants at 6 weeks, 6, and 24 months postpartum. VL measurements were performed at the Infectious Diseases Institute (IDI) Core Laboratory, Kampala, Uganda, a central laboratory certified by the American College of Pathologists. Nucleic acids were extracted and tested for HIV-1 RNA using the COBAS® AmpliPrep/ COBAS® TaqMan® HIV-1 Test procedure following the manufacturer's instructions (Roche Molecular Systems, Inc., Branchburg, NJ, 08876 USA) [24].

Viral suppression was defined as VL below 400 copies/mL. At the time of writing the study protocol, this limit was considered viral suppression by Uganda MOH [16]. Virological failure

was defined as a median viral load persistently above 1000 copies/ml for two consecutive viral load measurements within a three-month interval after at least 6 months of initiating ART. The final drug adherence variable at each time point was dichotomized as 'not adhering' if the viral load was ≥400 copies/ml irrespective of the level of self-reported adherence, or as 'adhering' if self-reported adherence in the last 3 days was ≧95% and viral load was <400 copies /ml.

Lastly, our secondary outcome of interest was HIV status and the HIV-free survival of children born to study participants at the 6-week and 18-month postpartum visits as per the National PMTCT program recommendation and practice.

## Statistical analyses

We estimated a target sample size of at least 270 women living with HIV needed to be enrolled in each of the study arms to provide 90% power at 5% significance level to detect a difference of 15% or more in adherence to visits and drugs or virological suppression between the intervention and control arms. Follow-up time was calculated in person-years, with each participant contributing time from enrolment to censoring at the date when a participant was lost to follow-up, died, or reached the end of the study. All the analyses were done using Stata Version 15.1 (Statacorp LP, College Station, Texas 77845 USA).

We compared participants' baseline characteristics in the FLC and SOC arms by providing the Standardized Mean Differences (SMD) with a threshold for assessing imbalance of 0.2. We performed an individually-randomized group treatment trial (IRGT) and accounted for clustering by peer groups in the intervention arm, and pseudo peer groups in the control arm, with each pseudo peer group made up of controls to peer group members in the intervention arm in the Kaplan Meier and Cox proportional hazard analyses.

The variable selection for the most appropriate model was determined using Akaike Information Criterion (AIC) which assessed the model that best fitted the data. We used the Wilcoxon–Breslow–Gehan test to assess the difference in HIV-free survival of participants' children between the FLC and SOC arms. Log-rank and Chi-Square p-values were used to test the equality of Kaplan-Meier survival probabilities and hazard rates for failure to retain in care for any reason by study arm. Cox Proportional Hazard regression model adjusted for unbalanced baseline variables was used to estimate the hazard rate ratio of failing to retain participants in care with 95% confidence intervals between the FLC and the SOC arms.

## Predictors of retention in care

Cox-proportional hazard regression models were also used to assess the effect measure of intervention on being retained in care.

The variable selection for the most appropriate model was determined using Akaike Information Criterion (AIC) which assessed the model that best fitted the data.

## Ethical and regulatory considerations

The study was approved by Uganda's National Council of Science and Technology (UNCST), the Joint Clinical Research Centre (JCRC) Institutional Review Board (IRB), as well as the IRBs of Johns Hopkins University (JHU) and the University of California San Francisco (UCSF). Written informed consent was obtained from all study participants by study counselors in the presence of an impartial witness before enrolment in the study. The authors confirm that all ongoing and related trials for this intervention are registered.

## Results

Between May 2016 and September 2017, a total of 1192 HIV-positive pregnant women were assessed for study eligibility. Of these, 45% (540/1192) were found eligible and randomized 1:1 to either the intervention or the control arm until each arm accrued 270 participants across all study sites (Fig 1).

### Baseline demographic characteristics

Table 1 provides the number of participants, clinic visits made, and person-years of follow-up by baseline socio-demographic characteristics of the study population stratified by study arm. These results show that despite randomization, there were no major imbalances between the two study arms at a SMD of 0.2.

### Adherence to PMTCT clinic appointments

There was no statistically significant difference between the intervention and control arms in the number of women who started or completed their postpartum visits at 6 weeks postpartum or in the first or second year of follow-up (Table 2).

### Retention in care

Overall, retention in care was high across both study arms with 83.0% of all participants remaining in care at the end of follow-up (Table 3). Significantly, more women were retained in care at the end of follow-up in the FLC arm (86.7%) compared to the SOC arm (79.3%, p = 0.0221). Also, more women in the SOC arm (n = 27, 10.0%) compared to the FLC arm (n = 12, 4.4%) were terminated before the end of follow-up due to relocation.

There was a significant difference between the control and intervention arms in the Kaplan-Meier survival functions to remain in care until the end of the study (24 months postpartum visit) (Wilcoxon-Breslow test for equality of survivor functions $Chi^2$ (1) = 11.94, p-value = 0.0005) (Fig 2).

### Adherence to ART

Table 4 shows that participants self-reported optimal adherence to taking >95% of their ART medication in the last 3 days at their 6 weeks, 6, and 24 months postpartum visits. There was no statistically significant difference in optimal adherence to ART between the arms. The median viral loads of participants at each of these time points were <100 copies/ml with overall decrease over time, but no statistically significant differences between the two arms.

Table 5 shows that the rate of dropout from care was approximately 2.4 times greater among participants randomized to the SOC arm versus the FLC arm (aHR = 2.363, 95% CI: 1.199–4.656, p = 0.013). Participants enrolled from rural health facilities and those aged 15–24 years were also significantly less likely to be retained in care compared to their urban (p = 0.015) or their older counterparts (p = 0.099) respectively. Educational level, gravidity, disclosure status, tribe, and income were considered but did not make it to the final model.

### HIV-free survival of children

Of 492 women with a visit at 18 months postpartum, 8 of their 431 (1.9%) infants tested positive for HIV. Most (7/8) of these infants had tested positive for HIV by the 6th-week postpartum visit. The proportion of HIV-positive children born to mothers enrolled in the FLC arm (1.4%) was not significantly different from that observed in the SOC arm (2.4%) (p = 0.472).

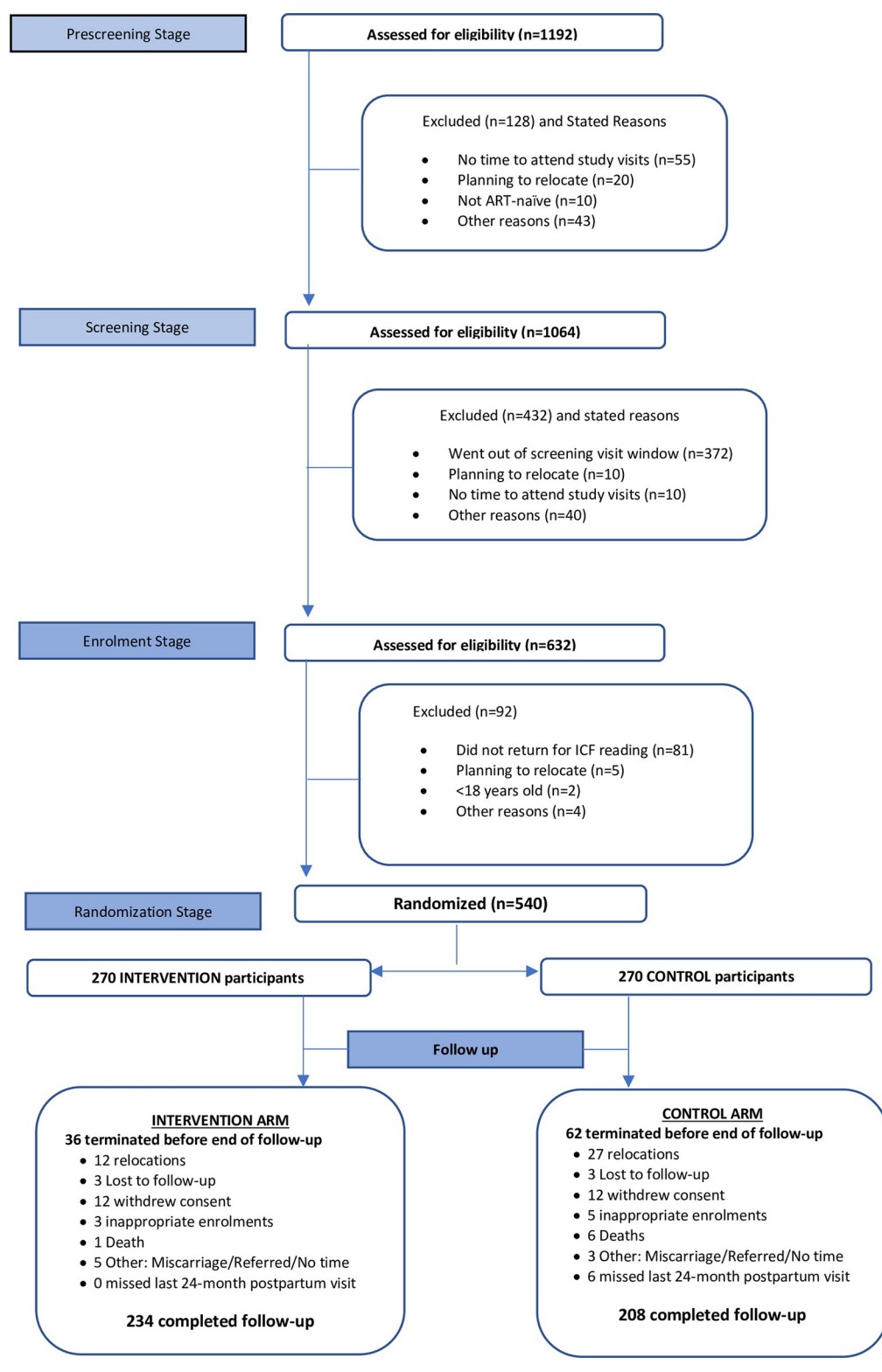

**Fig 1. CONSORT participant flow diagram.**

**Table 1. Baseline socio-demographic characteristics of the study population (N = 540).**

| Socio-demographic characteristics at baseline | Intervention (FLC Arm) | | | | Control (SOC Arm) | | | | Standardized Mean Difference |
|---|---|---|---|---|---|---|---|---|---|
| | Person-years of follow-up [PYFU] for ALL visits | | Total no. of women in FLC Arm | | Person-years of follow-up [PYFU] for ALL visits | | Total no. of women in SOC Arm | | |
| | Visits | PYFU | N | col % | Visits | PYFU | N | col % | *SMD |
| **Enrolment Sites** | | | | | | | | | 0.008 |
| Kampala-Urban | 2455 | 392.9 | 200 | 74.1 | 2271 | 354.1 | 201 | 74.4 | |
| Mityana-Rural | 907 | 144.5 | 70 | 25.9 | 912 | 142.8 | 69 | 25.6 | |
| **Marital Status** | | | | | | | | | 0.093 |
| Married/Co-habiting | 2607 | 418.2 | 209 | 77.4 | 2565 | 399.0 | 217 | 80.4 | |
| Never Married | 502 | 77.4 | 40 | 14.8 | 389 | 61.1 | 33 | 12.2 | |
| Separated/Divorced/Widowed | 253 | 41.8 | 21 | 7.8 | 229 | 36.8 | 20 | 7.4 | |
| **Age** | | | | | | | | | 0.198 |
| 18–24 years | 1415 | 222.9 | 116 | 43.0 | 1659 | 257.0 | 143 | 53.0 | |
| 25–34 years | 1654 | 267.7 | 130 | 48.2 | 1318 | 206.5 | 111 | 41.1 | |
| 35+ years | 293 | 46.8 | 24 | 8.9 | 286 | 33.3 | 16 | 5.9 | |
| **Educational level** | | | | | | | | | 0.085 |
| Tertiary | 194 | 30.9 | 16 | 5.9 | 116 | 17.9 | 9 | 3.3 | |
| Secondary | 1429 | 227.3 | 113 | 41.9 | 1810 | 282.4 | 145 | 53.7 | |
| Primary | 1611 | 260.2 | 131 | 48.5 | 1140 | 176.8 | 103 | 38.2 | |
| No education | 128 | 18.9 | 10 | 3.7 | 117 | 19.7 | 13 | 4.8 | |
| **Religion** | | | | | | | | | 0.062 |
| Catholic | 1185 | 188.0 | 93 | 34.4 | 1100 | 169.8 | 93 | 34.4 | |
| Protestant | 858 | 139.8 | 69 | 25.6 | 724 | 115.4 | 57 | 21.1 | |
| Moslem | 676 | 106.0 | 56 | 20.7 | 688 | 110.4 | 60 | 22.2 | |
| SDA/Pentecostal/others | 643 | 103.6 | 52 | 19.3 | 671 | 101.3 | 60 | 22.2 | |
| **Tribe** | | | | | | | | | 0.052 |
| Non-Ganda | 1495 | 240.7 | 123 | 45.6 | 1292 | 202.0 | 116 | 43.0 | |
| Ganda | 1867 | 296.6 | 147 | 54.4 | 1891 | 294.9 | 154 | 57.0 | |
| **Gravidity** | | | | | | | | | 0.179 |
| 1 | 613 | 96.1 | 51 | 18.9 | 630 | 96.6 | 54 | 20.0 | |
| 2 | 663 | 106.2 | 54 | 20.0 | 1011 | 159.7 | 84 | 31.1 | |
| 3 | 943 | 151.1 | 75 | 27.8 | 684 | 106.9 | 60 | 22.2 | |
| 4+ | 1143 | 184.0 | 90 | 33.3 | 858 | 133.7 | 72 | 26.7 | |
| **Disclosure to partner** | | | | | | | | | 0.075 |
| Yes | 1270 | 200.5 | 100 | 37.0 | 1396 | 214.6 | 118 | 43.7 | |
| No | 2036 | 328.1 | 165 | 61.1 | 1640 | 259.3 | 140 | 51.9 | |
| No partner | 56 | 8.8 | 5 | 1.9 | 147 | 23.0 | 12 | 4.4 | |
| **Disclosure to relatives** | | | | | | | | | 0.069 |
| Yes | 1333 | 212.6 | 106 | 39.3 | 1179 | 186.6 | 97 | 35.9 | |
| No | 2029 | 324.7 | 164 | 60.7 | 2004 | 310.3 | 173 | 64.1 | |
| **Monthly Income (USD)*** | | | | | | | | | 0.046 |
| 1.35–40.5 | 591 | 94.2 | 46 | 17.1 | 772 | 121.2 | 65 | 24.1 | |
| >40.5–135.1 | 946 | 149.5 | 75 | 27.8 | 627 | 96.5 | 51 | 18.9 | |
| >135.1 | 602 | 99.0 | 48 | 17.8 | 589 | 90.4 | 47 | 17.4 | |
| Don't know | 1223 | 194.7 | 101 | 37.4 | 1195 | 188.8 | 107 | 39.6 | |
| **TOTAL No. of Participants** | **3362** | **538.4** | **270** | **50.0** | **3183** | **497.2** | **270** | **50.0** | |
| TOTAL No. of All visits | | | 3362 | 51.4 | | | 3183 | 48.6 | |

*(Continued)*

**Table 1.** (Continued)

| | Intervention (FLC Arm) | | | Control (SOC Arm) | | | |
|---|---|---|---|---|---|---|---|
| No. of Scheduled PP visits | | | 2017 | 83.0 | | 1854 | 76.3 |

PP = postpartum

* SMD is Standardized Mean Diffirence with a threshold for assessing imbalance of > 0.2

† 1 USD = 3,650 UG Shillings (UGX) at the time of enrolment

PYFU = Person Years of Follow Up. Follow-up time was calculated in person-years, with each participant contributing time from enrolment to censoring at either: the date of death, or the date loss to follow-up was reported, or end of follow-up date (24-month postpartum visit)

By the 18-month postpartum visit, the HIV-free survival of children born to mothers in the FLC arm was 93.8% [95% CI: 89.9–96.2] compared to 83.0% [95% CI: 75.9–87.7] among children born to mothers in the SOC arm. The difference between the two arms was significant (p = 0.0031).

## Discussion

In this RCT, we found that there was no statistical difference through 24 months of postpartum follow-up in adherence to PMTCT clinic visits or to taking ART among pregnant and postpartum women started on Option B+ ART during pregnancy and randomized to the FLC intervention or the MOH SOC services. We however found that women in the FLC arm were statistically significantly more likely to have remained in PMTCT care through 2 years postpartum compared to the SOC arm. Furthermore, 1.9% of infants tested positive for HIV, however, there was no statistical difference in the proportion of HIV-positive children born to mothers enrolled in the FLC arm compared to those in the SOC arm. At 18 months, HIV-free survival of children born to mothers in the FLC arm was significantly different from children born to mothers in the SOC arm.

These results are comparable to the study by Masereka et al, which found that 87.9% of women were retained in care after being initiated on Option B+ in Uganda [25]. However, this study was not a randomized control trial and did not test peer support groups. Our data is also

**Table 2. Adherence to PMTCT clinic appointments.**

| Adherence to PMTCT clinic appointments<br>*Primary outcomes measure (A)* | At 6 Weeks Postpartum visit | | Number attended at least 4 quarterly visits in the 1st year postpartum measured at the 12 Months Postpartum visit | | Number attended at least 4 quarterly visits in the 2nd year postpartum measured at the 24 Months Postpartum visit | |
|---|---|---|---|---|---|---|
| | FLC ARM | SOC ARM | FLC ARM | SOC ARM | FLC ARM | SOC ARM |
| Intention To Treat analysis:<br>Includes all women who started the time window of interest (6 weeks postpartum, the first or the second year of postpartum follow-up) | 242/270 (89.6%) | 231/270 (85.6%) | 223/262 (85.1%) | 197/247 (79.8%) | 187/245 (76.3%) | 168/224 (75.0%) |
| Fisher's Exact p-values | P = 0.192 | | P = 0.129 | | P = 0.748 | |
| As treated analysis:<br>Includes all women who completed the time window of interest (6 weeks postpartum, the first or the second year of postpartum follow-up) | 242/262 (92.4%) | 231/247 (93.5%) | 223/245 (91.0%) | 197/224 (87.9%) | 187/229 (81.7%) | 168/207 (81.2%) |
| Fisher's Exact p-values | P = 0.730 | | P = 0.294 | | P = 0.903 | |

PP = postpartum

ᵃ Women who attended ≥ 4 quarterly visits in the 1st year PP

ᵇ Women who attended > 4 quarterly visits in the 2nd year PP

**Table 3. Retention in care by study arm at the end of the study.**

| Follow-up status at end of study | Study Arm, n (%) | | |
|---|---|---|---|
| | **FLC = 270** | **SOC = 270** | **Total = 540** |
| | **n (column %)** | **n (column %)** | **n (column %)** |
| **Terminated before end of follow-up** | **36 (13.3%)** | **56 (20.7%)** | **92 (17.0%)** |
| Terminated due to relocation | 12 (4.4%) | 27 (10.0%) | 39 (7.2%) |
| Lost to follow-up | 3 (1.1%) | 3 (1.1%) | 6 (1.1%) |
| Withdrawn from study | 15 (5.5%) | 17 (6.3%) | 32 (5.9%) |
| Termination due to death | 1 (0.4%) | 6 (2.2%) | 7 (1.3%) |
| Other—Miscarriage/Referred/No time | 2 (0.7%) | 3 (1.1%) | 5 (0.9%) |
| No information | 3 (1.1%) | 0 (0.0%) | 3 (0.6%) |
| **Missed last 24 months PP visit** | **0 (0.0%)** | **6 (2.2%)** | **6 (1.1%)** |
| **Completed follow-up** | **234 (86.7%)** | **208 (77.0%)** | **442 (81.9%)** |
| **Retention in care** | 1 - (36/270) **(86.7%)** | 1 - (56/270) **(79.3%)** | 1 - (92/540) **(83.0%)** |
| **p-value**[†] | **0.0221** | | |

n = Number of individuals; PP = Postpartum

[†] Chi-square (1) = 8.453

consistent with a study from Malawi which found that at 24 months after initiation of Option B+ ART, retention in the peer group models was 80% and 83% in the facility-based and community-based arms respectively, compared with 60% in the standard of care arm [17]. Taken together, these findings reinforce the concept that community-based peer group support may enhance retention in PMTCT care in Uganda and similar settings.

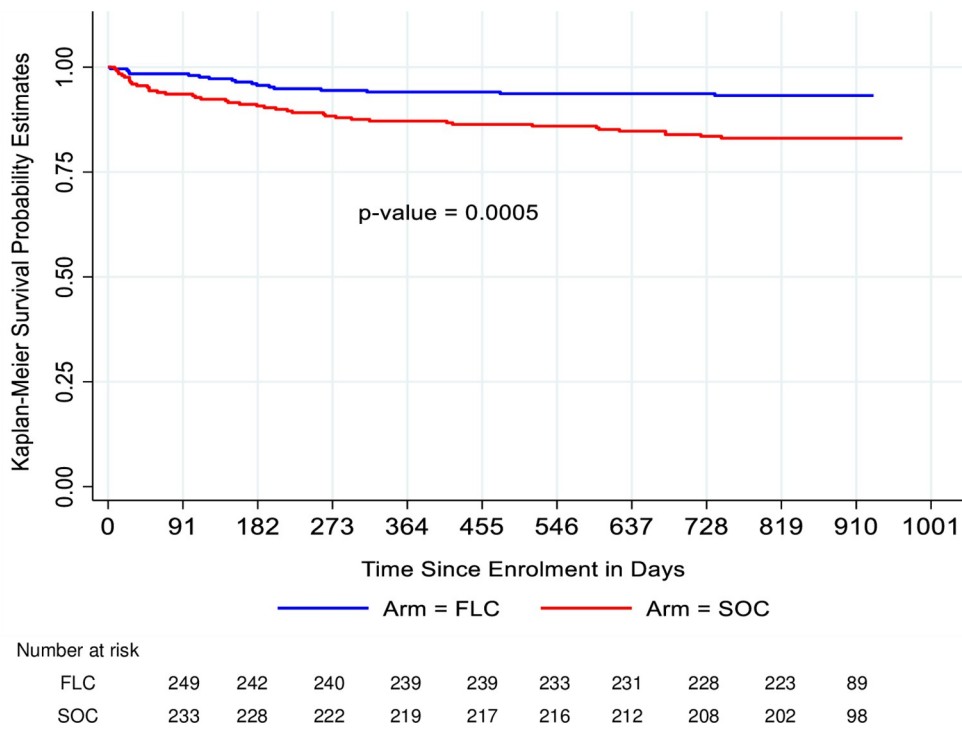

**Fig 2. Kaplan-Meier survival curves for retention in care up to 24 months postpartum.**

**Table 4. Adherence to Option B+ ART by study arm.**

| Adherence to Option B+ ART | 6 weeks postpartum | | 6 months postpartum | | 24 months postpartum | |
|---|---|---|---|---|---|---|
| **Self-reported** | **FLC** | **SOC** | **FLC** | **SOC** | **FLC** | **SOC** |
| Proportion of doses missed in past 3 days* | 11/242 **(4.6%)** | 9/232 **(3.9%)** | 29/225 **(12.9%)** | 34/216 **(15.7%)** | 22/223 **(9.9%)** | 20/200 **(10.0%)** |
| Adherence to taking ARV for option B+ ** | 1–4.6% **(95.4%)** | 1–3.9% **(96.1%)** | 1–12.9% **(87.1%)** | 1–15.7% **(84.3%)** | 1–9.9% **(90.1%)** | 1–10.0% **(90.0%)** |
| p-values | p = 0.718 | | p = 0.122 | | p = 0.644 | |
| **Viral load (VL)** | **FLC** | **SOC** | **FLC** | **SOC** | **FLC** | **SOC** |
| Median $\log_{10}$ VL (IQR) [†] | 1.80 | 1.86 | 1.64 | 1.52 | 1.30 | 1.30 |
| *Median copies/mL (IQR)* [†] | (1.30–2.61) | (1.34–2.78) | (1.30–2.66) | (1.30–3.79) | (1.30–2.77) | (1.30–2.83) |
| | *63 cp/ml* | *72 cp/ml* | *44 cp/ml* | *33 cp/ml* | *20 cp/ml* | *20 cp/ml* |
| | *(20–407)* | *(22–603)* | *(20–457)* | *(20–6,166)* | *(20–589)* | *(20–676)* |

\* No. of women with sub-optimal adherence in the past 3 days prior to clinic visit / Total number of women visiting the clinic and responding to the past 3 days adherence questions at each visit of interest (6 weeks pp, 6 months and 24 months pp). These results have accounted for the validation of self-reported adherence using viral load measures (Self-reported adherence with a viral load <400 copies/ml was classified as Adherent and self-reported adherence with a viral load ≥400 copies/ml was classified as non-adherent)

\*\* 1 minus the proportion of women self-reported as sub-optimally taking their medication in the past 3 days

[†] At 6 weeks postpartum, we used VL test results for samples taken at delivery.

In both arms of this RCT, 87% or more participants self-reported optimal adherence to taking their ART at 6 weeks, 6, and 24 months postpartum and the median viral load levels of participants were under 100 copies/ml at each of these time points with no significant difference between the FLC and SOC arms. We did not find fluctuations in longitudinal adherence during the postpartum period, although there was a drop during the follow up period.

A systematic review by Wubneh et al in Ethiopia found that the pooled estimate of adherence to Option B+ ART among 1852 pregnant and lactating women was 84% [26], which is comparable to our finding. Our finding is also consistent with a cohort study done in Malawi that found 90%-100% adherence to ART among women on Option B+ observed from 4 to 21 months postpartum [27]. That study used pharmacy records to measure adherence as opposed to self-reports and VL validation which may limit comparisons with our study.

Although we found high self-reported adherence to ART that corroborated with low median viral loads, we did not find significant differences in these outcomes between the FLC and SOC arms at any follow-up time points. The correlation between self reported adherence to ART and low median viral load echoes the findings of a study among 452 women on PMTCT care and treatment in South Africa where it was found that a raised viral load was consistently associated with lower median adherence scores [28]. However, the lack of a significant difference in adherence to PMTCT clinic visits or to ART or in median viral loads between FLC and SOC arms at

**Table 5. Multivariate Cox proportional hazard model for failure to remain in care.**

| Variables | | N = 540 | Adjusted Hazard Ratio (aHR) | [95% Conf. Interval] | p-value |
|---|---|---|---|---|---|
| **Study arm** | FLC | 270 | 1.000 | | |
| | SOC | 270 | 2.363 | 1.199–4.656 | 0.013 |
| **Urban status** | Urban | 401 | 1.000 | | |
| | Rural | 139 | 0.304 | 0.116–0.797 | 0.015 |
| **Age** | 15–24 years | 259 | 1.000 | | |
| | 25+ years | 281 | 0.526 | 0.245–1.128 | 0.099 |

Model selection using AIC and accounting for clustering by peer groups

any follow-up time points may result from a study effect or still a 'contamination' between the FLC and SOC arms among both study staff and participants in the study facilities [29].

Few children were found infected with HIV postnatally and most of them were likely infected *in utero* or at birth given the timing of their positive test result. However, our finding that HIV-free survival of children born to mothers in the FLC arm was significantly higher than those in the SOC arm is noteworthy as it implies that more children died of non-HIV causes in the latter arm. Many factors may have accounted for this outcome. Peer support and less variation in drug adherence among mothers in the FLC arm are some of the factors that may have contributed to this. Additionally, infant feeding counseling with emphasis on the importance of exclusive breastfeeding on demand and hygenic complementary feeding from 6 months onword could have contributed to this. Furthermore, increased economic security from money earned through IGAs provided the ability for mothers in the FLC arm to better feed and care for their babies [30–37].

## Limitations

Our study's main limitation is that our RCT was individually and not cluster randomized, which may have resulted in a study effect as well as cross-contamination between the FLC and the SOC arms in the same clinics over the long-term follow-up period. The non-blinding of randomization arms could have led to contamination when study staff attended both FLC and SOC participants in the same clinic, which could have positively influenced behaviours of participants in the SOC arm. Additionally, study participants in both FLC and SOC were given transport reimbursement for their study visits which may have positively influenced adherence to study visits. Lastly, the feasibility of our intervention would depend on its cost and cost-effectiveness, which we did not assess.

## Conclusions

Our study found that the FLC intervention significantly increased retention in care and HIV-free survival of children of pregnant mothers living with HIV through 24 months postpartum. These findings are encouraging and suggest that programmatic interventions such as the FLC may contribute to the goal of eliminating MTCT of HIV in Uganda. We recommend scaling up peer group support, community drug distribution, and wherever possible, sustainable income-generation intervention strategies in the context of PMTCT. The lack of a significant difference in adherence to PMTCT clinic visits or ART between the intervention and control arms requires more research to understand factors that contribute to adherence to PMTCT clinic visits or to ART among women in Option B+.

## Supporting information

**S1 Checklist. CONSORT 2010 checklist of information to include when reporting a randomised trial*.**
(DOCX)

**S1 Dataset.**
(XLSX)

## Acknowledgments

We would like to thank the FLC for Option B+ study team who were vital in the collection of this data and the study participants without whom this study would not have been possible.

## Author Contributions

**Conceptualization:** Samuel S. Malamba, Jaco Homsy, Rachel King, Clemensia Nakabiito, Zikulah Namukwaya, Mary Glenn Fowler, Philippa Musoke.

**Data curation:** Gordon Rukundo, Samuel S. Malamba, Jaco Homsy.

**Formal analysis:** Gordon Rukundo, Samuel S. Malamba.

**Investigation:** Alexander Amone, Grace Gabagaya, Priscilla Wavamunno, Joyce Namale-Matovu, Irene Lubega, Jaco Homsy, Rachel King, Clemensia Nakabiito, Zikulah Namuk-waya, Mary Glenn Fowler, Philippa Musoke.

**Methodology:** Alexander Amone, Grace Gabagaya, Priscilla Wavamunno, Joyce Namale-Matovu, Jaco Homsy, Mary Glenn Fowler, Philippa Musoke.

**Project administration:** Alexander Amone, Grace Gabagaya, Priscilla Wavamunno, Philippa Musoke.

**Resources:** Samuel S. Malamba, Philippa Musoke.

**Software:** Gordon Rukundo.

**Supervision:** Alexander Amone, Grace Gabagaya, Priscilla Wavamunno, Joyce Namale-Matovu, Zikulah Namukwaya, Philippa Musoke.

**Writing – original draft:** Alexander Amone, Grace Gabagaya, Jaco Homsy, Rachel King, Mary Glenn Fowler, Philippa Musoke.

**Writing – review & editing:** Alexander Amone, Grace Gabagaya, Priscilla Wavamunno, Joyce Namale-Matovu, Samuel S. Malamba, Irene Lubega, Jaco Homsy, Rachel King, Clemensia Nakabiito, Mary Glenn Fowler, Philippa Musoke.

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
