## [Decision Letter · Decision Letter 0]

24 Aug 2023

PONE-D-23-18840Enhanced Peer-Group Strategies to Support the Prevention of Mother-to-Child HIV transmission leads to increased retention in Care in Uganda: A Randomized controlled trialPLOS ONE

Dear Dr. Amone,

Thank you for submitting your manuscript to PLOS ONE. After careful consideration, we feel that it has merit but does not fully meet PLOS ONE’s publication criteria as it currently stands. Therefore, we invite you to submit a revised version of the manuscript that addresses the points raised during the review process.

We look forward to receiving your revised manuscript.

Kind regards,

Dorina Onoya

Academic Editor

PLOS ONE

Journal Requirements:

3. Thank you for providing the following Funding Statement: 

“PM received funding from NIH/Eunice Kennedy Shriver National Institute of Child Health and Human Development (NICHD) grant # IR01HD080476-01.”

We note that one or more of the authors is affiliated with the funding organization, indicating the funder may have had some role in the design, data collection, analysis or preparation of your manuscript for publication; in other words, the funder played an indirect role through the participation of the co-authors.

If the funding organization did not play a role in the study design, data collection and analysis, decision to publish, or preparation of the manuscript and only provided financial support in the form of authors' salaries and/or research materials, please review your statements relating to the author contributions, and ensure you have specifically and accurately indicated the role(s) that these authors had in your study in the Author Contributions section of the online submission form. Please make any necessary amendments directly within this section of the online submission form.  Please also update your Funding Statement to include the following statement: “The funder provided support in the form of salaries for authors [insert relevant initials], but did not have any additional role in the study design, data collection and analysis, decision to publish, or preparation of the manuscript. The specific roles of these authors are articulated in the ‘author contributions’ section.”

If the funding organization did have an additional role, please state and explain that role within your Funding Statement.

Please also provide an updated Competing Interests Statement declaring this commercial affiliation along with any other relevant declarations relating to employment, consultancy, patents, products in development, or marketed products, etc. 

“The Friends for Life Circles for Option B+ study which was funded by NIH/Eunice Kennedy Shriver National Institute of Child Health and Human Development (NICHD) grant # IR01HD080476-01.”

5. Please include your tables as part of your main manuscript and remove the individual files. Please note that supplementary tables (should remain/ be uploaded) as separate "supporting information" files"

Reviewers' comments:

Reviewer's Responses to Questions

**Comments to the Author**

1. Is the manuscript technically sound, and do the data support the conclusions?

Reviewer #1: Partly

Reviewer #2: Partly

2. Has the statistical analysis been performed appropriately and rigorously? 

Reviewer #1: No

Reviewer #2: I Don't Know

3. Have the authors made all data underlying the findings in their manuscript fully available?

Reviewer #1: Yes

Reviewer #2: Yes

4. Is the manuscript presented in an intelligible fashion and written in standard English?

Reviewer #1: Yes

Reviewer #2: Yes

5. Review Comments to the Author

Reviewer #1: General comments:

From a statistical methods standpoint, my main question regards whether clustering needs to be accounted for in these analyses. In the discussion (lines 350-351), you note that the limitation that the study was not cluster-randomized. I agree with this, though I would argue that you are performing an individually-randomized group treatment trial (IGRT) since the intervention is delivered in groups despite the participants being individually randomized. IGRTs are important when the group-based delivery of the intervention can have an effect on outcomes. Though, this can also be true if the intervention is delivered to groups who may be correlated in some way. For some background on IGRTs, I suggest reading this article https://ajph.aphapublications.org/doi/full/10.2105/AJPH.2007.127027. Sometimes it is true the group dynamic may not induce any correlation, but I think since the peer groups were defined geographically, there is a greater chance of correlation in groups.

IGRTs with a clustered intervention but non-clustered control can be challenging to analyze since statistical procedures need groups defined in both arms. In this instance, since your intervention groups were determined geographically, I think the control groups can be defined in the same way.

Failure to account for clustering in analyses can result in type I errors, so I see this change as necessary to ensure proper inferences are made. That change is not possible in Fisher's exact test, but I think that approach should be changed (see specific comment #2). There are approaches to account for clustered data in Kaplan Meier but without knowing which software package was used, I cannot recommend anything. Finally, Cox models have methods which account for clustered data in most software packages.

Specific comments:

1. (line 119) What were the possible block sizes?

2. (lines 218-219, 245-248) Significance testing between intervention groups is generally frowned upon because a non-significant p-value does not indicate that groups are the same. For info on the topic in relation to baseline imbalance in randomized trials see Altman, https://doi.org/10.2307/2987510 and Senn, https://doi.org/10.1002/sim.4780131703. My usual recommendation is to remove the significance testing from tables like this and use standardized mean differences (SMD) to assess imbalance (see Austin, https://doi.org/10.1080/03610910902859574). Most often, I see authors using an SMD of 0.1 or 0.2 as a threshold for assessing imbalance, though this could vary by field.

3. Please indicate the software package and version used for these analyses.

4. (lines 284-286) Variable selection methods should be included in the statistical methods section.

5. (lines 284-286) Sun et al. (http://dx.doi.org/10.1016/0895-4356(96)00025-X) found that bivariate screening can miss a variable that may be a confounder even when a p-value higher than what you have used. Bivariate screening can be considered a form of stepwise variable selection, which usually do not do a good job of finding the most appropriate model (e.g., https://doi.org/10.1002/sim.3943). Generally it's better to select based on more robust criteria. I prefer shrinkage-based models, such as lasso, for variable selection. At a minimum, I would encourage using measures which assess the fit of the model. AIC or BIC may be possible, for which you could you Burnham and Anderson (doi: 10.1177/0049124104268644) as a guide.

Reviewer #2: Review Plos One

This is a randomized controlled trial among women with HIV at urban and rural health facilities in Uganda. The trial recruited 540 women and women were randomized to a peer-group intervention (FLC) or standard of care (SOC). Outcomes of interest were 1. adherence to PMTCT at 6 weeks, 12 and 24 months, 2. self-reported adherence to ART at 6 weeks, 6 and 24 months, and 3. Retention in care at 24 months, 4. HIV status and HIV-free survival of infants at 18 months of age. This study reported an effect of the FLC intervention on retention in care at end of follow-up, HIV-diagnoses among infants, and HIV-free survival among infants. However, no effect as seen for adherence to PMTCT and ART. The findings of this trial are interesting and important implying that peer-group session could increase retention in care and increase survival among their infants. My major concern is that it is not clear what the prespecified primary and secondary aims were of this study. Was this a pre-registered trial? Moreover, the abstract, methods and discussion needs more elaboration on the design of the study, key findings, discuss them in context to other previous research and what the implication future research is.

Major comments:

Abstract

- Suggest clarifying when women were recruited? Various timepoints during pregnancy?

- Suggest clarifying the primary and secondary outcomes of the trial investigated in this particular study.

- Provide more information on the FLC intervention, when was it introduced, how often did the participants meet, when did the intervention end.

Methods

- Please clarify if the trial was non-blinded

- Pease clarify if the trials was preregistered, and the prespecified primary and secondary endpoints of this trial.

Discussion

- Lines 309-314: Suggest to provide a summary of all results. Including HIV status and HIV-free survival.

- Lines 315-316, was this a randomized trial? Did this study also test a peer-group intervention?

- Lines 319-321: Please clarify what authors are referring to. Currently other studies that seem to see an effect are stated. However, authors are then stating that these results are explaining the null-finding.

- Lines 345-348: Why would the FLC arm have better economic security? Elaborate more.

- Lines: 350-353: I would think that the non-blinding is more of an issue then the individual-based. Cluster randomized also have their weaknesses.

- Lines: 353-355: Were also women in the SOC arm given transport reimbursement?

- Conclusion: Also include the null findings for your primary outcome and future aspects of this.

Table 5

- Please clarify if the HR for the intervention effect was adjusted or not.

Minor comments:

Abstract

- Please add that retention in care was an outcome of interest to the methods as results for them are shown and conclusions are being made.

- Please add results for HIV status of HIV-free survival of infant as this this is an outcome stated in the methods section.

Introduction

- Lines 79-86. Please clarify the design of these studies.

- Lines 87-92. Long sentence, suggest to split up into two to make it easier for the reader to follow.

Methods

- Line 112: Were women scheduled back to study clinic if eligible?

- 110-114: Were only women who were newly diagnosed included in the study? Please clarify.

- Lines 219-222. This is not part of the primary aim, but a prespecified secondary aims? Recommend mentioning in the outcome section.

- Page 9-11: please clarify the prespecified primary outcomes of the trial and secondary outcomes.

-

Results

- Line 258 says 83% retained in care whereas lines 268-269 says 87 % retained in care by the end of follow-up.

- Lines 283-286: I would suggest move this paragraph to the methods section

- Line 298-299, repeating of “tested”

Discussion

- Line 335: It is unclear with wat authors mean by “close correlation”

- Lines 338-340: Please elaborate more one what is meant by “the lack of more distinct outcome”

6. PLOS authors have the option to publish the peer review history of their article (what does this mean?). If published, this will include your full peer review and any attached files.

Reviewer #1: No

Reviewer #2: No

---

## [Author Response · Author response to Decision Letter 0]

11 Nov 2023

22 October 2022

To the Editor-in-Chief

PLOS ONE Journal

Dear Editor-in-Chief, 

We would like to submit our amended manuscript titled: “Enhanced Peer-Group strategies to support prevention of Mother-to-Child HIV transmission leads to increased retention in care in Uganda: A Randomized controlled trial" following reviewers’ comments received on 25 Aug 2023. We have provided point-by-point responses to the editor and reviewers’ comments below.

We thank you for your interest in this manuscript and look forward to being considered for publication in the PLOS ONE Journal. Please address all correspondence concerning this manuscript to the corresponding author at aamone@mujhu.org.

We would also like to amend the Funding statement “PM received funding from NIH/Eunice Kennedy Shriver National Institute of Child Health and Human Development (NICHD) grant # IR01HD080476-01.” Kindly update this to read, “The funders had no role in study design, data collection and analysis, decision to publish, or preparation of the manuscript.”

Sincerely,

Alexander Amone

On behalf of all the authors

Please note: All page and line numbers refer to the clean view of the tracked changes version of the last submitted version of the manuscript.

Number Reviewer comments Author response 

 Journal requirements 

1. Please ensure that your manuscript meets PLOS ONE's style requirements, including those for file naming. The PLOS ONE style templates

 We have followed the PLOS ONE’s style requirements, and the files were named accordingly.

2. We note that the grant information you provided in the ‘Funding Information’ and ‘Financial Disclosure’ sections do not match. When you resubmit, please ensure that you provide the correct grant numbers for the awards you received for your study in the ‘Funding Information’ section We have provided the correct grant number under the “funding information” section. 

3. Thank you for providing the following Funding Statement: 

“PM received funding from NIH/Eunice Kennedy Shriver National Institute of Child Health and Human Development (NICHD) grant # IR01HD080476-01.”

We note that one or more of the authors is affiliated with the funding organization, indicating the funder may have had some role in the design, data collection, analysis or preparation of your manuscript for publication; in other words, the funder played an indirect role through the participation of the co-authors The information “PM received funding from NIH/Eunice Kennedy Shriver National Institute of Child Health and Human Development (NICHD) grant # IR01HD080476-01” was provided in error. This was the funding for implementation of this clinical trial. 

We have clarified this in the cover letter. No author is affiliated with the funding organisation. 

Kindly update this information to read, “The funders had no role in study design, data collection and analysis, decision to publish, or preparation of the manuscript.”

“The Friends for Life Circles for Option B+ study which was funded by NIH/Eunice Kennedy Shriver National Institute of Child Health and Human Development (NICHD) grant # IR01HD080476-01.”

Please include your amended statements within your cover letter; we will change the online submission form on your behalf Funding information “The Friends for Life Circles for Option B+ study which was funded by NIH/Eunice Kennedy Shriver National Institute of Child Health and Human Development (NICHD) grant # IR01HD080476-01” has been removed from the manuscript as advised by the reviewer.

We would like to maintain the correct funding information “The funders had no role in study design, data collection and analysis, decision to publish, or preparation of the manuscript.” Therefore, kindly update the funding information in the online submission to reflect this correct funding statement.

5. Please include your tables as part of your main manuscript and remove the individual files. Please note that supplementary tables (should remain/ be uploaded) as separate "supporting information" files"

 All tables have been included in the main manuscript as advised by the reviewer.

Review comments to the Author 

1 Reviewer # 1: General comments:

From a statistical methods standpoint, my main question regards whether clustering needs to be accounted for in these analyses. In the discussion (lines 350-351), you note that the limitation that the study was not cluster-randomized. I agree with this, though I would argue that you are performing an individually-randomized group treatment trial (IGRT) since the intervention is delivered in groups despite the participants being individually randomized. IGRTs are important when the group-based delivery of the intervention can have an effect on outcomes. Though, this can also be true if the intervention is delivered to groups who may be correlated in some way. For some background on IGRTs, I suggest reading this article. Sometimes it is true the group dynamic may not induce any correlation, but I think since the peer groups were defined geographically, there is a greater chance of correlation in groups.

IGRTs with a clustered intervention but non-clustered control can be challenging to analyze since statistical procedures need groups defined in both arms. In this instance, since your intervention groups were determined geographically, I think the control groups can be defined in the same way.

Failure to account for clustering in analyses can result in type I errors, so I see this change as necessary to ensure proper inferences are made. That change is not possible in Fisher's exact test, but I think that approach should be changed (see specific comment #2). There are approaches to account for clustered data in Kaplan Meier but without knowing which software package was used, I cannot recommend anything. Finally, Cox models have methods which account for clustered data in most software packages.

 We have added to the statistical analysis section the statement: We performed an individually-randomized group treatment trial (IRGT) and accounted for clustering by peer groups in the intervention arm, and pseudo peer groups in the control arm, with each pseudo peer group made up of controls to peer group members in the intervention arm in the Kaplan Meier and Cox proportional hazard analyses

Lines 235-239

 Specific comments 

1 (line 119) What were the possible block sizes? We have added the randomization block sizes that ranged between 2-8.

Line 132

2 (lines 218-219, 245-248) Significance testing between intervention groups is generally frowned upon because a non-significant p-value does not indicate that groups are the same. For info on the topic in relation to baseline imbalance in randomized trials. My usual recommendation is to remove the significance testing from tables like this and use standardized mean differences (SMD) to assess imbalance. Most often, I see authors using an SMD of 0.1 or 0.2 as a threshold for assessing imbalance, though this could vary by field.

 We have removed the significance testing from Table 1 and instead provided the Standardized Mean Differences (SMD) with a threshold for assessing imbalance of 0.2.

Lines 234-235

3 Please indicate the software package and version used for these analyses. We have included the software package and version used for analysis as follows: Stata Version 15.1 (Statacorp LP, College Station, Texas 77845 USA). 

Lines 232-233

4 (lines 284-286) Sun et al. found that bivariate screening can miss a variable that may be a confounder even when a p-value higher than what you have used. Bivariate screening can be considered a form of stepwise variable selection, which usually do not do a good job of finding the most appropriate model. Generally it's better to select based on more robust criteria. I prefer shrinkage-based models, such as lasso, for variable selection. At a minimum, I would encourage using measures which assess the fit of the model. AIC or BIC may be possible, for which you could you Burnham and Anderson (doi: 10.1177/0049124104268644) as a guide.

 We have stated in the statistical methods section that “The variable selection for the most appropriate model was determined using Akaike Information Criterion (AIC) which assessed the model that best fitted the data”

Lines 240-241

Reviewer 2 Review Plos One 

 Abstract 

1 Suggest clarifying when women were recruited? Various time points during pregnancy? We have clarified when women were recruited in the study to read “pregnant women living with HIV at urban and rural health facilities in Uganda were enrolled in the study at any gestation age.” 

Line 39

2 Suggest clarifying the primary and secondary outcomes of the trial investigated. We have clarified the primary and secondary outcomes of the trial investigated.

Line 35

3 Provide more information on the FLC intervention, when was it introduced, how often did the participants meet, when did the intervention end.

 We have provided more information on the FLC intervention to include when it was introduced, frequency of meetings and when the intervention ended to read: “The FLC groups were formed within 4 months after enrollment, were followed up until the last participant from the group reached 24 months post delivery and exited from the study as a group.” 

Lines 44-46

4 Methods 

Please clarify if the trial was non-blinded The trial was non-blinded and this has been clarified in the methods section.

 Line 102

5 Pease clarify if the trials was preregistered, and the prespecified primary and secondary endpoints of this trial.

 The trial was registered on ClinicalTrials.gov and this has been included in the methods section.

Lines 111-112.

The primary and secondary end points have been clarified.

Lines 118-224

6 Discussion

Lines 309-314: Suggest to provide a summary of all results. Including HIV status and HIV-free survival. As suggested by the reviewer, we have provided a summary of all results, including HIV status and HIV-free infant survival. 

Lines 385-389

7 Lines 315-316, was this a randomized trial? Did this study also test a peer-group intervention?

 This was not a randomised trial and has been clarified to read “However, this study was not a randomized control trial and did not test peer support groups, like the current study.” Lines 391-392

8 Lines 319-321: Please clarify what authors are referring to. Currently other studies that seem to see an effect are stated. However, authors are then stating that these results are explaining the null-finding. As suggested by the reviewer, we have clarified this to read, “Taken together, these findings reinforce the concept that community-based peer group support may enhance retention in PMTCT care in Uganda and similar settings.”

Lines 396-397

9 Lines 345-348: Why would the FLC arm have better economic security? Elaborate more.

 We have elaborated this to read “Peer support and less variation in drug adherence among mothers in the FLC arm are some of the factors that may have contributed to this. Furthermore, increased economic security from money earned through IGAs provided the ability for mothers in the FLC arm to better feed and care for their babies and these are supported by an ample literature.”

Lines 423-426

10 Lines: 350-353: I would think that the non-blinding is more of an issue then the individual-based. Cluster randomized also have their weaknesses. We have clarified this to read, “Our study's main limitation is that our RCT was individually and not cluster randomized, which may have resulted in a study effect as well as cross-contamination between the FLC and the SOC arms in the same clinics over the long-term follow-up period. The non-blinding of randomization arms could have led to contamination when study staff attended both FLC and SOC participants in the same clinic, which could have positively influenced behaviours of participants in the SOC arm.”

Line 429-434

11 Suggest clarifying when women were recruited? Various timepoints during pregnancy?

 We have clarified that “Pregnant women were recruited at any gestation age.”

Lines 114

12 Suggest clarifying the primary and secondary outcomes of the trial investigated in this particular study.

 The primary and secondary outcomes have been clarified.

Lines 202, 222-223

13 Lines: 353-355: Were also women in the SOC arm given transport reimbursement?

 Women in the SOC were also given transport reimbursement and the statement has been modified to read “Additionally, study participants in both FLC and SOC were given transport reimbursement for their study visits which may have positively influenced adherence to study visits.”

Lines 434-437

14 Conclusion: 

Also include the null findings for your primary outcome and future aspects of this. We have included the null findings in the primary outcome as suggested by the reviewer

Line 444-447

 Table 5

- Please clarify if the HR for the intervention effect was adjusted or not. We have clarified table 5 to state that the HR is adjusted.

Line 363

 Minor comments 

 Please add that retention in care was an outcome of interest to the methods as results for them are shown and conclusions are being made. We have added that retention in care was an outcome of interest to the methods section.

Line 190

 Please add results for HIV status of HIV-free survival of infant as this this is an outcome stated in the methods section. Results for the HIV free survival of infants have been added in the outcome section.

 Introduction

 Lines 79-86. Please clarify the design of these studies.

 We have clarified the design of these studies cited to read “A scoping literature review of studies done between 2000-2020 identified barriers to taking ART among pregnant women that included financial constraints limiting access to food and transport, and side effects of the therapy.”

Line 81

 Lines 87-92. Long sentence, suggest to split up into two to make it easier for the reader to follow.

 As suggested by the reviewer, we have split the sentence in to two parts, making it easier to read. 

Lines 97-102

 Methods

 Line 112: Were women scheduled back to study clinic if eligible?

 Yes, we have clarified this to read “If eligible, they were scheduled to return to the study clinics for consent procedures within one month of starting once daily Option B+ ART.” 

Lines 120-121

 110-114: Were only women who were newly diagnosed included in the study? Please clarify.

 We have clarified this to read “Pregnant women newly diagonised HIV-positive by testing through the routine National PMTCT Program and starting Option B+ were referred to study counselors by the clinic PMTCT counselors for prescreening,” Line 121-123

 Lines 219-222. This is not part of the primary aim, but a prespecified secondary aims? Recommend mentioning in the outcome section. The statement “Maternal adherence to at least 4 scheduled PMTCT clinic appointments at the end of each year of postpartum follow-up, adherence to ART, and retention in care, as well as the HIV status of children, were also summarized using proportions and chi-square p-values” has been removed from the paper as it was repeated.

 Lines 283-286: I would suggest move this paragraph to the methods section

 As suggested by the reviewer, this paragraph has been moved to the methods section. Line 249-250

 Page 9-11: please clarify the prespecified primary outcomes of the trial and secondary outcomes.

 We have clarified the primary outcomes as “The primary outcomes of interest included adherence to PMTCT clinic appointments at 6 weeks, 12, and 24 months postpartum, adherence to Option B+ ART at 6 weeks, and 6 and 24 months postpartum, and retention in care at the end of study follow up.” 

Lines 118-224

 Results

 Line 258 says 83% retained in care whereas lines 268-269 says 87 % retained in care by the end of follow-up.

 The statement “Overall, the estimated proportion of participants retained in care was high, with 87.0% of all participants remaining in care until the end of the study” has been removed. 

Line 336 

 Line 298-299, repeating of “tested”

 The repeat word “tested” has been deleted to read “Of 492 women with a visit at 18 months postpartum, 8 of their 431 (1.9%) infants tested positive for HIV.” 

Line 369

 Discussion

 Line 335: It is unclear with what authors mean by “close correlation”

 We have clarified this to read in part, “The correlation between self reported adherence to ART and low median viral load…”

Lines 411-412

 Lines 338-340: Please elaborate more one what is meant by “the lack of more distinct outcome”

 This has been clarified to read as follows:

“the lack of a significant difference in adherence to PMTCT clinic visits or to ART or in median viral loads between FLC and SOC arms at any follow-up time points may result from a study effect or still a ‘contamination’ between the FLC and SOC arms among both study staff and participants in the study facilities”. 

Lines 414-418

---

## [Decision Letter · Decision Letter 1]

10 Jan 2024

Enhanced Peer-Group Strategies to Support the Prevention of Mother-to-Child HIV transmission leads to increased retention in Care in Uganda: A Randomized controlled trial

PONE-D-23-18840R1

Dear Dr. Amone,

We’re pleased to inform you that your manuscript has been judged scientifically suitable for publication and will be formally accepted for publication once it meets all outstanding technical requirements.

Kind regards,

Dorina Onoya

Academic Editor

PLOS ONE

Reviewers' comments:

Reviewer's Responses to Questions

**Comments to the Author**

1. If the authors have adequately addressed your comments raised in a previous round of review and you feel that this manuscript is now acceptable for publication, you may indicate that here to bypass the “Comments to the Author” section, enter your conflict of interest statement in the “Confidential to Editor” section, and submit your "Accept" recommendation.

Reviewer #1: All comments have been addressed

Reviewer #3: All comments have been addressed

2. Is the manuscript technically sound, and do the data support the conclusions?

Reviewer #1: (No Response)

Reviewer #3: Yes

3. Has the statistical analysis been performed appropriately and rigorously? 

Reviewer #1: (No Response)

Reviewer #3: Yes

4. Have the authors made all data underlying the findings in their manuscript fully available?

Reviewer #1: (No Response)

Reviewer #3: Yes

5. Is the manuscript presented in an intelligible fashion and written in standard English?

Reviewer #1: (No Response)

Reviewer #3: Yes

6. Review Comments to the Author

Reviewer #1: (No Response)

Reviewer #3: the paper is surely of high quality bringing solution to the hard issue of retention in care for EMTCT

My few comments are listed below

-ABSTRACT :

line 39 : can you specify moment of randomization line 39

Line 42 : could you change the expression "validated" by plasma viral load measurements as it is not the gold standard for measurement of adherence. You can suppress validated and put ...."and "

line 45 : not clear with 4 months enrollment ; question does FLC groups formed during pregnancy prior delivery or prior 6 weeks post partum

METHODS

Line 156 does group counseling contains infant feeding issues ; if yes specify

RESULTS

Figure one Trial profile Can timing of each stage be added including randomization

DISCUSSION

line 422 to 426 the main point to highlight and discuss is HIV free survival. the statement of authors on timing of MTCT in utero or intra partum can be supported by the fact that women were starting option B+ during pregnancy.

But a known factor affecting HIV free survival is infant feeding practice. So not only assuming that FLC arm better feed and care for their babies, it should be good to have idea of infant feeding counseling in this arm as previous stated in methodology.

7. PLOS authors have the option to publish the peer review history of their article (what does this mean?). If published, this will include your full peer review and any attached files.

Reviewer #1: No

Reviewer #3: **Yes: **ANNE ESTHER NJOM NLEND

---

## [Editor Report · Acceptance letter]

26 Mar 2024

PONE-D-23-18840R1 

PLOS ONE

Dear Dr. Amone, 

I'm pleased to inform you that your manuscript has been deemed suitable for publication in PLOS ONE. Congratulations! Your manuscript is now being handed over to our production team.

Kind regards, 

on behalf of

Dr. Dorina Onoya 

Academic Editor

PLOS ONE